# Identification of Structural Variation from NGS-Based Non-Invasive Prenatal Testing

**DOI:** 10.3390/ijms20184403

**Published:** 2019-09-07

**Authors:** Ondrej Pös, Jaroslav Budis, Zuzana Kubiritova, Marcel Kucharik, Frantisek Duris, Jan Radvanszky, Tomas Szemes

**Affiliations:** 1Faculty of Natural Sciences, Comenius University, 841 04 Bratislava, Slovakia (Z.K.) (J.R.) (T.S.); 2Geneton Ltd., 841 04 Bratislava, Slovakia (J.B.) (M.K.) (F.D.); 3Comenius University Science Park, 841 04 Bratislava, Slovakia; 4Slovak Center of Scientific and Technical Information, 811 04 Bratislava, Slovakia; 5Institute for Clinical and Translational Research, Biomedical Research Center, Slovak Academy of Sciences, 845 05 Bratislava, Slovakia

**Keywords:** copy number variants, next generation sequencing, non-invasive prenatal testing, population study

## Abstract

Copy number variants (CNVs) are an important type of human genome variation, which play a significant role in evolution contribute to population diversity and human genetic diseases. In recent years, next generation sequencing has become a valuable tool for clinical diagnostics and to provide sensitive and accurate approaches for detecting CNVs. In our previous work, we described a non-invasive prenatal test (NIPT) based on low-coverage massively parallel whole-genome sequencing of total plasma DNA for detection of CNV aberrations ≥600 kbp. We reanalyzed NIPT genomic data from 5018 patients to evaluate CNV aberrations in the Slovak population. Our analysis of autosomal chromosomes identified 225 maternal CNVs (47 deletions; 178 duplications) ranging from 600 to 7820 kbp. According to the ClinVar database, 137 CNVs (60.89%) were fully overlapping with previously annotated variants, 66 CNVs (29.33%) were in partial overlap, and 22 CNVs (9.78%) did not overlap with any previously described variant. Identified variants were further classified with the AnnotSV method. In summary, we identified 129 likely benign variants, 13 variants of uncertain significance, and 83 likely pathogenic variants. In this study, we use NIPT as a valuable source of population specific data. Our results suggest the utility of genomic data from commercial CNV analysis test as background for a population study.

## 1. Introduction

Copy number variation (CNV) is a segment of DNA with length ≥1 kbp which is presented at a variable copy number in comparison to the reference genome. CNVs include insertions, deletions and duplications, which result in copy number gain or copy number loss [1]. It was shown that CNVs are important cause of structural variations in the human genome [2]. Research of the past decades revealed that these variations are functionally and evolutionary significant and contribute to the population diversity and human genetic diseases [3,4].

Various methods for CNV detection have been developed, from the conventional cytogenetic analysis (e.g., G-banded karyotype) through microarray-based methods (e.g., comparative genomic hybridization) to next-generation sequencing (NGS) [5]. Genomic microarrays provide a genome-wide coverage at a much higher resolution than a conventional cytogenetic analysis. This is the reason why microarray-based methods have been standard for CNV detection [6,7]. However, this method has limited resolution, accuracy, and several other limitations are noted in the literature [8]. In recent years, NGS has become a valuable tool for clinical diagnostics and to provide sensitive and accurate approaches for detecting genomic variations, e.g., CNVs. With the reducing cost of this method, numbers of NGS based CNV detection tests is increasing [9,10].

In our previous study, we described non-invasive prenatal test (NIPT) based on analysis of plasma DNA from pregnant women [11,12,13]. This test uses low-coverage massively parallel sequencing of whole-genome for detection of CNV aberrations [14]. With the informed consent of these patients we generated an amount of credible genomic data from thousands of pregnant women. Since these patients represent a relatively standard sample of local female population, we hypothesized this data could be used not only for primary purpose as prenatal screening but also as a valuable source of data for population study. The objective of the present study is based on our previous work which suggests the use of NIPT as a valuable source of population specific allelic frequencies [15].

## 2. Results

We obtained CNV profile for 22 autosomes from 5018 pregnant women (Figure 1). Together, we identified 225 CNVs ranging from 600 kbp to 7820 kbp with median size 820 kbp (Appendix A). These variants include 178 duplications (79.11%) and 47 deletions (20.89%) with median size 830 kbp for duplications and 800 kbp for deletions. As can be seen, the majority of identified CNVs were approximately 600–700 kbp long (Figure 2a). Most variants (28) were found on the chromosome 2, while on the chromosome 15 we detected only one variant. We did not identify any deletions on chromosomes 11, 15, 20, and 22 (Figure 2b). The identified CNVs came from 212 individuals, corresponding to frequency 4.2% of CNV ≥ 600 kbp in our cohort. The vast majority of individuals (95.28%) displayed a single CNV; only 4.72% exhibited more than one variant. The most frequently detected variant was the CNV duplication in chromosome location 2p22 with a total of 11 detection events; however, the frequency of every CNV was calculated as less than 1%, thus all variants were considered to be rare. The largest CNV was duplication spanning 7820 kbp in chromosome location 10q21.1.

Variants were compared with ClinVar database records and following results were obtained. Together, 137 CNVs (60.89%) were overlapping with previously described variants in full extent, 66 CNVs (29.33%) were partially overlapping and 22 CNVs (9.78%) did not overlap with any previously described variant according to ClinVar. Some of our CNVs overlap with variants previously observed among patients with pathogenic phenotypes, e.g., developmental delay, intellectual disability, etc. (Table 1).

The identified variants were classified based on criteria in AnnotSV database [19]. In summary, we identified 129 likely benign variants, 13 variants of uncertain significance and 83 likely pathogenic variants. According to AnnotSV, 207 CNVs overlap with known genes and only 18 CNVs were localized in non-coding areas. Regarding the type of CNV, we identified approximately 3.8 times more CNV gains than CNV losses. These variants were more frequently present in non-coding regions; however, duplications overlap coding regions nearly 6.4 times more frequently than the deletions (Table 2).

## 3. Discussion

Knowledge of population genetic studies, e.g., Human Genome Project, has changed genomics and had tremendous impact on current medicine [20,21]. Detection of CNVs within and between populations is important to understand the plasticity of our genome and to elucidate its possible contribution to disease management [22]. Based on these statements, we are suggesting the additional utility of genomic data generated through routine NIPT screening based on low-coverage massively parallel whole-genome sequencing of total plasma DNA from pregnant women. This test provides a lot of credible genomic data that can be used as background for population studies. Our results show that 4.2% of individuals carry CNV ≥ 600 kbp, suggesting a relatively high frequency of large CNVs in the Slovak population. These findings are consistent with results from Cooper et al., which presented one of the largest studies investigating the role of rare CNVs in intellectual disability and developmental delay, analyzing data from 15,767 affected individuals and 8329 controls. They showed that 25.7% of affected individuals and 11.5% of controls harbor CNVs > 400 kbp [23].

Overall, there were approximately four times higher frequency of duplications compared to deletions (Table 2). The underrepresentation of deletions is consistent with previous reports, where large deletions were less common than large duplications when considering CNVs > 500 kbp [24,25]. These results are concordant with the hypothesis that CNV losses are more deleterious [26]. All variants together span 238.52 Mbp; however, only 3.71 Mbp (1.56%) were identified in coding regions. These 3.71 Mbp were spread through 207 CNVs (92%) overlapping with coding sequences. Since the gene density is calculated at 5–23 genes per Mbp [27], there is a low probability that a CNV ≥ 600 kbp will occur exactly in the non-coding region. Therefore, we expected most CNVs of such length to be at least partially overlapping the coding regions. We have shown that duplications affect coding regions approximately two times more frequently than deletions (1.71% vs 0.93% for duplications and deletions, respectively). Sudmant et al. also found that duplications and deletions exhibit fundamentally different population-genetic properties. Duplications are subjected to weaker selective constraint, hence affect genes four times more likely than deletions, indicating that they provide a larger target for adaptive selection [3].

Clinically relevant CNVs can be found in databases such as ClinVar, DECIPHER, ECARUCA and the International Standards for Cytogenomic Arrays Database. When we compared our results with ClinVar database, we found at least 22 variants (17 CNV gains; 5 CNV losses) in regions without any previous record (Appendix A). For example, we have identified a CNV loss on the chromosome location 3q26.3 that is present consistently in three of our samples, but it was not previously described in the database. This deletion encompassing approximately one half of sequence from the 3′ end of a gene N-acetylated alpha-linked acidic dipeptidase-like 2 (*NAALADL2*). It has been shown that deletions involving *NAALADL2* are found in the general population [28]. On a closer view, we found that our largest duplication in chromosome 10q21.1 overlaps the complete sequence of gene Protocadherin Related 15 (*PCDH15*). Duplications in this gene have been shown to be associated with Usher syndrome type 1 (OMIM: # 601067), which is characterized by deafness, vestibular areflexia, and prepubertal onset of retinitis pigmentosa [29,30]. Although the NIPT enables the detection of maternal CNVs, current analyses do not interpret these findings. Maternal aberrations can be clinically actionable or potentially harmful for the fetus. Brison et al. suggest reporting these variants if clinically relevant because it can improve pregnancy management and promote the health of the fetus or the mother or both [31]. On the other hand, the identification and reporting of such CNVs represent a big challenge for genetic counselors; thus, further guidelines to improve patient counseling are needed [32]. It is also known that performing NIPT may incidentally lead to the diagnosis of maternal malignancy. Giles et al. showed, 80% of genetic counselors recognized it would be beneficial in the future to use NIPT for neoplasm screening, however, more than 90% affirmed that guidelines are necessary to better prepare for these cases [33].

Performing large numbers of parental samples is expensive, but the need for parental testing will diminish by accumulating data about benign CNVs [16]. Recently, an updated, higher resolution map of CNVs that are not associated with adverse phenotypes, based on 55 studies, was developed. Zarrei et al. estimated that up to 9.5% of the genome contributes to CNV. Additionally, they have found approximately 100 genes that can be homozygously deleted without producing apparent phenotypic consequences. This map is a great contribution to the interpretation of new CNV findings, for clinical and research applications [34]. As clinical laboratories adopt CNV analysis, these resources will become invaluable for the clinician to discriminate pathogenic from non-disease associated CNVs [8]. However, there is still a need for appropriate recommendations or guidelines related to evaluation of CNV findings and for their classifications. The main limitation of our study remains the size of detected CNVs; however, with improving laboratory and computational methods, as well as lowering the cost of sequencing, this limit should decrease. Currently, our method was validated to CNVs with minimal length 600 kbp, while the vast majority of CNVs are smaller than 500 kbp [35]. On the other hand, CNVs larger than 500 kbp are strongly associated with morbid consequences such as developmental disorders and cancer [22]. Despite mentioned limitation, we showed, NIPT may be utilized for the identification of common structural variations in population, and it could contribute to the interpretation of CNV findings in clinical research.

## 4. Materials and Methods 

In our previous work we described non-invasive prenatal test (NIPT) based on low-coverage (0.3×) massively parallel whole-genome sequencing of total plasma DNA for detection of CNV aberrations longer than 600 kbp [14]. This test generates amount of credible genomic data, from thousands of pregnant women which represent a relatively standard sample of local population. We reanalyzed NIPT genomic data from 5018 patients to calculate frequencies of CNV aberrations in the Slovak population. All subjects gave their informed consent for inclusion before they participated in the study. Informed consent includes permission to process the sample for further analysis maintaining the anonymity but does not include a statement for contacting the patient again in case of a clinically significant maternal finding. Therefore, it was possible to use samples processed in the past, but due to anonymization we were not able to contact the patients and associate the finding with the phenotype. The study has been approved by the Ethical Committee of the Bratislava Self-Governing Region (Sabinovska ul.16, 820 05 Bratislava) on 30 April 2015 under the decision ID 03899_2015.

### 4.1. Sample Preparation and Sequencing

Blood from pregnant women was collected into EDTA tubes and kept at 4 °C temperature until plasma separation. Blood plasma was separated within 36  h after collection and stored at −20 °C until DNA isolation. DNA was isolated using Qiagen DNA Blood Mini kit (QIAGEN, Hilden, Germany). Standard fragment libraries for massively parallel sequencing were prepared from isolated DNA using an Illumina TruSeq Nano kit (Illumina, San Diego, CA, USA) and a modified protocol described previously [11]. Briefly, to decrease laboratory costs, we used reduced volumes of reagents, which was compensated by nine cycles of PCR instead of eight as per protocol. Physical size selection of cfDNA fragments was performed using specific volumes of magnetic beads in order to enrich fetal fraction. Illumina NextSeq 500/550 High Output Kit v2 (75 cycles) (Illumina, San Diego, CA, USA) was used for massively parallel sequencing of prepared libraries using pair-end sequencing with read length of 2 × 35 bp on an Illumina NextSeq 500 platform.

### 4.2. Mapping and Read Count Correction

Sequencing reads were aligned to the human reference genome (hg19) using Bowtie 2 algorithm [36]. NextSeq-produced fastq files (two per sample; R1 and R2) were directly mapped using the Bowtie 2 algorithm with very-sensitive option. Next, for each sample, the unique reads were processed to eliminate the GC bias according to [37] with the exclusion of intrarun normalization. Briefly, for each sample the number of unique reads from each 20 kbp bin on each chromosome was counted. With empty bins filtered out, the locally weighted scatterplot smoothing (LOESS) regression was used to predict the expected read count for each bin based on its GC content. The LOESS-corrected read count for a particular bin was then calculated as RC = RC − ∣∣RC− RC∣∣⁠, where RC is the global average of read counts through all bins; RC is the fitted read count of that bin, and RC is its observed read count. PCA normalization has been further carried out to remove higher-order population artifacts on autosomal chromosomes [38,39]. At first, bin counts are transformed into a principal space. The first component represents the highest variability across individuals in the control set. To normalize the sample, bin counts corresponding to predefined number of top components are removed to reduce common noise in euploid samples. Bins without sufficient coverage that correspond to the low complexity genomic regions were excluded from the analysis. 

### 4.3. Segment Identification and CNV Calling

Normalized bin counts were analyzed by circular binary segmentation (CBS) algorithm provided by the R package DNAcopy (Seshan VE, Olshen A. DNAcopy: DNA copy number data analysis. R package version 1.48.0. 2016.) to identify same-coverage segments. CBS partitions a chromosome into regions with equal copy numbers. Segments longer than 600 kbp with abnormal copy number (at least 60% gain or loss of a single chromosomal segment) were marked as maternal and annotated using AnnotSV tool [40] and ClinVar database [41]. 

### 4.4. Data Processing

All computational steps were executed using Snakemake workflow engine [42]. Evaluation of maternal calls and generation of plots were performed using in-house Python scripts.

## 5. Conclusions

CNVs represent an important source of variations in the human genome. They are functionally and evolutionary significant and contribute to the population diversity and human genetic diseases. As NGS has become a valuable tool in research and in clinical settings, the number of NGS based tests has increased. Among them, CNV detection tests are also increasing. In this study, we confirmed our hypothesis and demonstrated that NIPT can be used also for the identification of common structural variations in population.

## Figures and Tables

**Figure 1 ijms-20-04403-f001:**
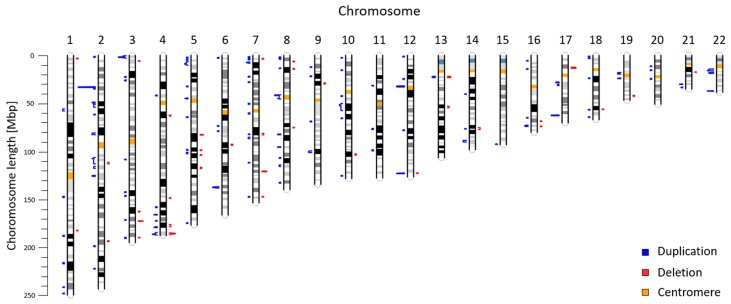
Chromosomal location of maternal CNVs identified by NIPT. The length of blue (duplication) and red (deletion) bars corresponds to the frequency of CNV ranging from minimum of 1 to maximum of 11 detections.

**Figure 2 ijms-20-04403-f002:**
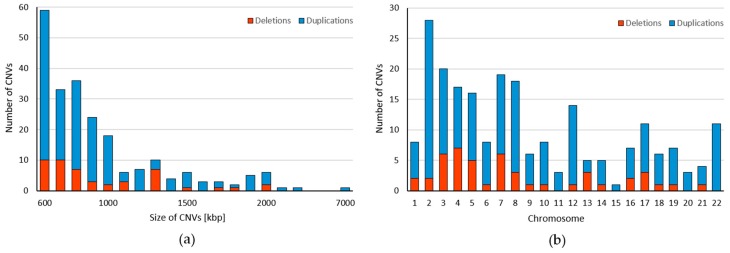
Characteristics of maternal CNVs identified from the NIPT. (**a**) Size distribution of detected CNVs ranging from ≥600 kbp to ≥7000 kbp. (**b**) Genomic distribution of CNV deletions (red) and duplications (blue) ≥600 kbp in Slovak population within the chromosomes.

**Table 1 ijms-20-04403-t001:** Variants overlapping with CNVs that were previously observed among patients with pathogenic phenotypes. Data acquired from the ClinVar database.

Variant Type	Location	Identifier	Phenotype	Events	Reference
Duplication	1q21.1-21.2	dbVar: nsv531885	Developmental delay AND/OR other significant developmental or morphological phenotypes, Global developmental delay	1	[16,17]
Duplication	2q33.1	OMIM: 609728.0002	Autosomal Recessive Spastic Ataxia with Leukoencephalopathy	1	[18]
Duplication	7q11.23	dbVar: nsv532240	Encephalopathy, Global developmental delay, Muscular hypotonia	1	[17]
Deletion	13q12.12	dbVar: nsv491643	Developmental delay AND/OR other significant developmental or morphological phenotypes, Seizures, Intellectual disability, Intrauterine growth retardation	2	[16]
Duplication	17q12	dbVar: nsv2775541	Developmental delay AND/OR other significant developmental or morphological phenotypes, Behavioral abnormality	1	[16,17]
Duplication	22q11.21	dbVar: nssv577068 nsv530653	Global developmental delay	3	[16,17]
Duplication	22q11.21	dbVar: nssv578923 nsv531796	Developmental delay AND/OR other significant developmental or morphological phenotypes	1	[16,17]
Duplication	22q11.23	dbVar: nssv13653977 nsv2769497	Short stature, Macrocephalus, Abnormality of the face, Intellectual disability	2	[16]

**Table 2 ijms-20-04403-t002:** Data shows number of identified CNVs sorted by the type of variant and number of Mega base pairs (Mbp) attributed to specific genomic location.

Type of Variant	Number of CNVs	Total Sequence (Mbp)	Coding Regions (Mbp)	Non-Coding Regions (Mbp)
CNV gain	178	191.54	3.27	188.27
CNV loss	47	46.98	0.44	46.54
Sum	225	238.52	3.71	234.81

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
