# Peer review of "Identification of Structural Variation from NGS-Based Non-Invasive Prenatal Testing"

_ijms, 2019, doi:10.3390/ijms20184403_

Round 1

Reviewer 1 Report

This is an interesting paper in which the authors have re-analyzed NIPT results to evaluate the frequency of maternal copy number variants. 

The study parameters need to be better defined.  Specifically, did the IRB and consent process include reporting to the patient any abnormal results seen in them?  Did the authors have any access to clinical information about the women tested?   A key question is how often a clinically significant CNV was consistent with the phenotype of the women and how often the information was confirming a previous diagnosis.  Similarly for potentially pathogenic findings. The authors should include in the Discussion some material about whether they think looking for maternal imbalances could be justified as part of prenatal testing.  This would seem to be fraught with ethical challenges but could be justified if it leads to better pregnancy management, diagnosis in the fetus (especially for variably expressed CNVs), or future risk management.  For the pathogenic CNVs in Table 1, how many of each type were there?   Was the very large imbalance on chromosome 10 also present in the fetus?

Author Response

Response to Reviewer 1 Comments

This is an interesting paper in which the authors have re-analyzed NIPT results to evaluate the frequency of maternal copy number variants.

Point 1: The study parameters need to be better defined.  Specifically, did the IRB and consent process include reporting to the patient any abnormal results seen in them?  Did the authors have any access to clinical information about the women tested?   A key question is how often a clinically significant CNV was consistent with the phenotype of the women and how often the information was confirming a previous diagnosis.  Similarly for potentially pathogenic findings. 

Response 1: Consent process did not include reporting of any abnormal results to the patients. Unfortunately, we do not have any access to clinical information about the women tested, therefore we could not compare CNV results with the phenotype of women. This information has been added to the materials and methods section of the manuscript.

Point 2: The authors should include in the Discussion some material about whether they think looking for maternal imbalances could be justified as part of prenatal testing.  This would seem to be fraught with ethical challenges but could be justified if it leads to better pregnancy management, diagnosis in the fetus (especially for variably expressed CNVs), or future risk management. 

Response 2: We included a paragraph on the challenging issue of identifying and reporting of maternal CNVs. The clinical significance of variants, pregnancy management and requirements for the guidelines were discussed.

Point 3: For the pathogenic CNVs in Table 1, how many of each type were there?

Response 3: We provided additional data (the number of variants of each type) to the Table 1 of the manuscript.

Point 4: Was the very large imbalance on chromosome 10 also present in the fetus? 

Response 4: We have not validated the resolution of the detection method to distinguish between maternal and fetomaternal signal yet. We therefore cannot say with certainty, however the amount of abundant reads that corresponds to 87% mosaic of the third copy, strongly suggests that the fetus is not affected.

Reviewer 2 Report

Excellent. No changes suggested.

Reviewer 3 Report

The reviewed manuscript represents a genomic population study, describing the ocurrence and identity of copy number variants in more than 5000 Slovak women. The text is well-written, based on sound methodology and provides numerical results that are mostly in accordance with current knowledge, although some aspects may differ a bit (chromosomal distribution, size of variants, ratio of deletions to duplications etc.) and be of interest to doctors and scientists studying heritable conditions associated with CNVs. Some of the most original parts of this work is the discovery of new variants not present in the ClinVar database and determination of the proportion of variants that are a likely cause of an abnormal condition in the patient.

I only have one bigger concern about reproducibility and a few specific comments that could possibly increase the quality of the paper.

1) In terms of reproducibility, the manuscript mentions in house Python scripts, a Snakemake workflow and overall does not provide enough information to reproduce the analysis on similar data. I suggest the authors critically re-read the Methods section and add information, either in text, as supplementary files or as references to stable online sources/data that would make it easier for someone to run a similar analysis. Scripts used to generate figures can be excluded, I suppose.

2) Low coverage sequencing is mentioned but I could not find a number for what the average coverage of the sequencing experiment was.

3) The results of 207 overlapping coding v. 18 in non-coding - what would be the number if 225 intervals with same size distribution would cover the genome randomly. In other words, it would be beneficial to state here how unexpected or expected these numbers were.

4) Discussion mentions 1.56% of deletions in coding regions - it is not immediately clear if this refers to the study or the work cited in the preceding sentence as reference number 25. 

Linguistic and semantic comments:

- "the NGS" <- perhaps definite article not needed here?
- "as it is seen" <- "as can be seen" sounds better to me
- citing the numbers 830 and 800 kbp using the word "respectively" is hard to follow for the reader, I suggest the authors write something like "830 for duplications and 800 for deletions".
- the majority of identified CNVs was <- I would suggest "were" here, since the meaning is "most of", not a specific kind of majority
- in full extend <- extent
- "showed that xx% of affected individuals and yy% of controls harboring" <- either "were harboring" or "harbor" or drop "that"

Author Response

This manuscript is a resubmission of an earlier submission. The following is a list of the peer review reports and author responses from that submission.